

# Serum BDNF levels and the antidepressant effects of electroconvulsive therapy with ketamine anaesthesia: a preliminary study

Wei Zheng[1,*], Qiaomei Cen[1,*], Sha Nie[1], Minyi Li[1], Rong Zeng[1], Sumiao Zhou[1], Dongbin Cai[2], Miaoling Jiang[1] and Xiong Huang[1]

[1] The Affiliated Brain Hospital of Guangzhou Medical University, Guangzhou, China
[2] Shenzhen Traditional Chinese Medicine Hospital, Shenzhen, China
* These authors contributed equally to this work.

Corresponding authors
Miaoling Jiang,
miaoling2011_0318@163.com
Xiong Huang, 1195768576@qq.com

## ABSTRACT

**Objective:** To firstly examine the relationship between serum brain-derived neurotrophic factor (BDNF) levels and antidepressant response to ketamine as an anaesthesia in electroconvulsive therapy (ECT) in Chinese patients with treatment-refractory depression (TRD).
**Methods:** Thirty patients with TRD were enrolled and underwent eight ECT sessions with ketamine anaesthesia (0.8 mg/kg) alone. Depression severity, response and remission were evaluated using the 17-item Hamilton Depression Rating Scale (HAMD-17). Enzyme-linked immunosorbent assay (ELISA) was applied to examine serum BDNF levels in patients with TRD at baseline and after the second, fourth and eighth ECT sessions. Baseline serum samples were also collected for 30 healthy controls.
**Results:** No significant differences were observed in serum BDNF levels between patients with TRD and healthy controls at baseline ($p > 0.05$). The remission rate was 76.7% (23/30) after the last ECT treatment, although all patients with TRD obtained antidepressant response criteria. Serum BDNF levels were not altered compared to baseline, even between remitters and nonremitters (all $p > 0.05$), despite the significant reduction in HAMD-17 and Brief Psychiatric Rating Scale (BPRS) scores after ECT with ketamine anaesthesia (all $p < 0.05$). The antidepressant effects of ECT with ketamine anaesthesia were not correlated with changes in serum BDNF levels (all $p > 0.05$).
**Conclusion:** This preliminary study indicated that serum BDNF levels do not appear to be a reliable biomarker to determine the antidepressant effects of ketamine as an anaesthesia in ECT for patients with TRD. Further studies with larger sample sizes are warranted to confirm these findings.

## INTRODUCTION

Electroconvulsive therapy (ECT) is widely considered to be the most effective nonpharmacological therapy for mental disorders (*Grover et al., 2018*; *Zong et al., 2020*), especially for major depressive disorder (MDD), despite negative public perceptions

(*Dean & Keshavan, 2017*; *Gajaria & Ravindran, 2018*; *Sackeim et al., 2007*). For example, *Petrides et al. (2001)* reported that the remission rate was 87% for both psychotic and nonpsychotic patients with MDD after an acute ECT course. To minimize the clinical risks and subjective unpleasantness during ECT, patients are administered an intravenous anaesthetic, such as thiopental, methohexital, propofol, ketamine or even a combination of ketamine and propofol (ketofol) (*Huang et al., 2020*; *Zheng et al., 2019a*). A recent meta-analysis of 16 trials ($n = 928$) found that ketamine used in ECT accelerated the improvement of depressive symptoms in patients with MDD, with a short-term advantage in antidepressive effect at the early stages of ECT (*Ren et al., 2018*).

Ketamine, an N-methyl-d-aspartate receptor (NMDAR) antagonist, has been widely used as an analgesic, anaesthetic and antihyperalgesic agent (*Radvansky et al., 2015*). Interestingly, a single ketamine infusion at sub-anaesthesia doses elicited a rapid but time-limited antidepressant effect in treatment-refractory depression (TRD) (*Hu et al., 2016*; *Zarate et al., 2006*). Repeated ketamine infusions at sub-anaesthesia doses have a cumulative and sustained antidepressant effect on TRD (*Phillips et al., 2019*; *Zheng et al., 2018*, *2019b*). Thus, ketamine used as an anaesthesia in ECT may enhance the antidepressant effects of ECT, while also having rapid independent antidepressant properties itself (*Erdil et al., 2015*; *Kranaster et al., 2011*; *Okamoto et al., 2010*; *Ren et al., 2018*; *Zheng et al., 2019a*). *Zhong et al. (2016)* found that ketamine anaesthesia achieved earlier antidepressant efficacy and a higher rate of remission than propofol anaesthesia and ketofol anaesthesia in TRD treated with ECT, suggesting that ketamine alone in ECT may represent an optimized therapy for TRD.

When compared to healthy controls, patients with TRD have low serum brain-derived neurotrophic factor (BDNF) levels (*Molendijk et al., 2014*; *Polyakova et al., 2015b*). Many, but not all studies (*Brunoni, Lopes & Fregni, 2008*; *Groves, 2007*; *Polyakova et al., 2015b*) found that low serum BDNF levels in patients with TRD were normalized after obtaining an antidepressant response. A few studies have examined the correlation of blood BDNF levels and the antidepressant response to ECT with ketamine anaesthesia or ketofol anaesthesia, but with inconsistent findings. For instance, a recent study reported that ECT treatment with ketamine anaesthesia, but not with methohexital anaesthesia, significantly increased plasma BDNF levels (*Carspecken et al., 2018*). However, another study found that ECT with ketofol anaesthesia was not correlated with enhanced serum BDNF levels (*Huang et al., 2020*). Moreover, baseline serum BDNF levels did not appear to be an eligible biomarker for predicting the antidepressant response to ECT with ketofol anaesthesia, as measured using the 17-item Hamilton Depression Rating Scale (HAMD-17) (*Huang et al., 2020*).

To date, no study has been conducted to examine whether serum BDNF levels can predict the rapid antidepressant response to ketamine used alone as an anaesthesia during ECT in Chinese patients with TRD. In this study, thirty Chinese patients with TRD were administered ketamine alone as an anaesthesia in ECT. We hypothesized that (1) patients with TRD would exhibit lower serum BDNF levels than healthy controls and (2) the antidepressant response to ketamine anaesthesia in ECT would be correlated with serum BDNF levels.
## METHODS

The protocol for this study was approved by the ethics review board of the Affiliated Brain Hospital of Guangzhou Medical University (Ethics Number: (2013)020). This study was conducted following the Declaration of Helsinki and was performed between February 2013 and December 2013.

### Participants

All patients were recruited from the wards of the Affiliated Brain Hospital of Guangzhou Medical University based on the following criteria: (1) age from 18 to 65 years; (2) diagnosed with major depression according to the International Statistical Classification of Diseases and Related Health Problems, 10th revision (ICD-10) with a current major depressive episode; (3) having severe depressive symptoms (HAMD-17 scores ≥ 20) at screening; (4) did not respond adequately to appropriate courses of at least two antidepressants in the current episode (*Huang et al., 2020*; *Zheng et al., 2020*); (5) had no history of severe physical illness (i.e., Parkinson disease) and no drug or alcohol abuse; (6) had no history of seizures; (7) was not breastfeeding or pregnant; and (8) had no any contraindication for ketamine anaesthesia and ECT. All patients with TRD provided written informed consent at the beginning of participation.

Thirty age and sex-matched apparently healthy volunteers were recruited from the local community during the same phase with no alcohol or other substance abuse/dependence or serious physical diseases. All healthy volunteers provided written informed consent at the beginning of participation.

### Treatment

All patients with TRD received eight ECT treatments with ketamine anaesthesia alone (0.8 mg/kg). ECT treatment was performed three times per week for three consecutive weeks for a total of eight treatments. During the courses of ECT, no psychiatric medications were prescribed to the subjects. The seizure threshold of each patient was determined based on the half-age method (% energy = half the age) (*Petrides & Fink, 1996*; *Yasuda et al., 2015*), and bitemporal ECT for each case was conducted by using Thymatron® IV device (Somatics LLC, Lake Bluff, IL, USA). Vital signs, such as temperature and heart rate, were regularly recorded.

### Clinical assessment

The HAMD-17 (*Hamilton, 1960*) and the Brief Psychiatric Rating Scale (BPRS) were used to assess the severity of depressive and psychotic symptoms, respectively, at baseline, after the second, fourth, and eighth sessions of ECT. Antidepressant response was defined as a 50% or greater reduction in HAMD-17 scores (*Lin & Lin, 2019*) and antidepressant remission as HAMD-17 scores ≤ 6 (*Riedel et al., 2010*). *Riedel et al. (2010)* found that the HAMD-17 cut-off ≤ 6 (area under the curve = 0.90) was correlated with a maximum sensitivity and specificity for defining remission criteria.

**Table 1 Comparison of demographic and clinical characteristics between patients with TRD and healthy controls.**

| Variables | Patients with TRD (n = 30) | | Healthy controls (n = 30) | | Statistics | | |
|---|---|---|---|---|---|---|---|
| | n | % | n | % | $\chi^2$ | df | P |
| Male | 14 | 46.7 | 13 | 43.3 | 0.07 | 1 | 0.795 |
| Employed | 24 | 80.0 | – | – | – | – | – |
| Married | 20 | 66.7 | – | – | – | – | – |
| Antidepressant response after the last ECT[a] | 30 | 100.0 | – | – | – | – | – |
| Antidepressant remission after the last ECT[b] | 23 | 76.7 | – | – | – | – | – |
| | Mean | SD | Mean | SD | t | df | P |
| Age (years) | 32.1 | 9.9 | 30.8 | 8.2 | 0.58 | 58 | 0.561 |
| Baseline serum BDNF levels (ng/ml) | 23.7 | 8.9 | 24.2 | 6.1 | −0.24 | 58 | 0.810 |

Notes:
[a] Antidepressant response was defined as a 50% or greater reduction in HAMD-17 scores.
[b] Antidepressant remission was defined as HAMD-17 scores ≤6 as recommended by Riedel et al.'s study.
Abbreviations: BDNF, brain-derived neurotrophic factor; df, degrees of freedom; ECT, electroconvulsive therapy; SD, standard deviation; TRD, treatment-refractory depression.

## Serum BDNF levels

Blood sample were collected from healthy controls at baseline. Blood samples from patients with TRD were collected at baseline and after the second, fourth, and eighth treatments of ECT. Before analysis of serum BDNF levels, blood samples were stored at −80 °C. In line with the manufacturer's instructions, an enzyme-linked immunosorbent assay (ELISA) kit (BDNF Emax Immunoassay System, Promega, USA) was used to examine serum BDNF levels. Absorbance at 450 nm wavelength was used to analyse BDNF concentrations based on a standard curve.

## Data analyses

Comparison of demographics and clinical characteristics between healthy controls and patients with TRD, and between remitters and nonremitters defined as HAMD-17 scores ≤ 6 (*Riedel et al., 2010*), was examined using the Student's *t*-test, Mann–Whitney *U*-test, chi-square test, or Fisher's exact test, as appropriate. A linear mixed model was used to compare serum BDNF levels and the severity of depressive and psychotic symptoms between remitters and nonremitters following eight ECT treatments. The correlation between changes in illness severity and changes in serum BDNF levels was analysed using Pearson's bivariate correlation analysis. All outcomes were analysed using IBM SPSS statistics version 25.0, and $p < 0.05$ was considered significant.

## RESULTS

### Participant characteristics

As shown in Table 1 and Fig. S1, no significant differences were observed in baseline serum BDNF levels between healthy controls and patients with TRD ($p > 0.05$). Similarly, no significant differences in terms of baseline serum BDNF levels, HAMD-17 scores or BPRS scores were found between remitters and nonremitters (all $p > 0.05$, Table 2).

**Table 2 Comparison of demographic and clinical characteristics between remitters and nonremitters after ECT.**

| Variables | Total samples (n = 30) | | Remitters (n = 23) | | Nonremitters (n = 7) | | Statistics | | |
|---|---|---|---|---|---|---|---|---|---|
| | n | % | n | % | n | % | $\chi^2$ | df | P |
| Male | 14 | 46.6 | 10 | 43.4 | 4 | 57.1 | 0.4 | 1 | 0.526 |
| Employed | 24 | 80.0 | 19 | 82.6 | 5 | 71.4 | —[a] | —[a] | 0.603 |
| Married | 20 | 66.7 | 14 | 60.8 | 6 | 85.7 | —[a] | —[a] | 0.372 |
| | Mean | SD | Mean | SD | Mean | SD | t/z | df | P |
| Age (years) | 32.1 | 9.9 | 31.2 | 10.5 | 35.3 | 7.5 | 0.9 | 28 | 0.344 |
| Baseline serum BDNF levels (ng/ml) | 23.7 | 8.9 | 24.3 | 9.8 | 21.8 | 5.8 | −0.6 | 28 | 0.543 |
| Baseline HAMD-17 scores | 26.7 | 1.6 | 26.7 | 1.7 | 26.8 | 1.1 | 0.3 | 28 | 0.769 |
| Baseline BPRS scores | 35.5 | 4.2 | 35.6 | 4.5 | 35.0 | 3.1 | −0.3 | 28 | 0.741 |

Notes:
[a] Fisher's exact test.
Abbreviations: BDNF, brain-derived neurotrophic factor; BPRS, the Brief Psychiatric Rating Scale; df, degrees of freedom; ECT, electroconvulsive therapy; HAMD-17, the 17-item Hamilton Depression Rating Scale ; SD, standard deviation.

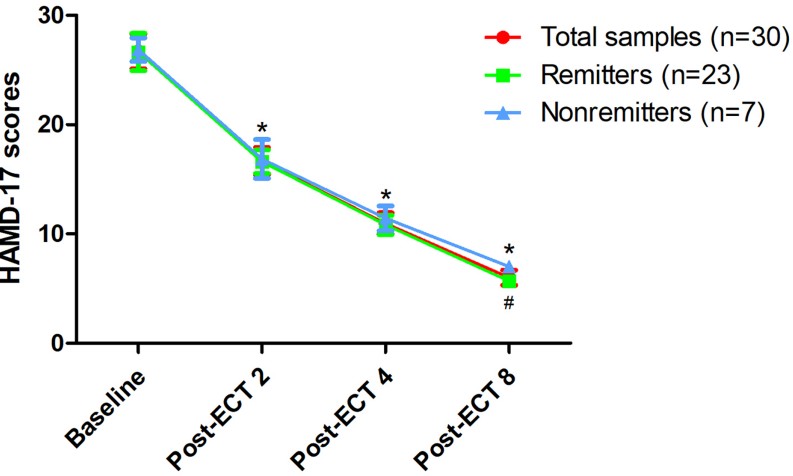

**Figure 1 Changes in depressive symptoms following eight ECT treatments.** *Significant difference was found at any of the indicated times when compared baseline ($p < 0.05$). #Significant difference was found between remitters and nonremitters at indicated times ($p < 0.05$). Abbreviations: ECT, electroconvulsive therapy; HAMD-17, the 17-item Hamilton Depression Rating Scale.

## Treatment remission and serum BDNF levels

After the last ECT treatment, the remission rate was 76.7% (23/30), while the response rate was 100% (30/30) (Table 1). Significant reduction in illness severity as measured by HAMD-17 (Fig. 1) and BPRS (Fig. S2) was observed following eight ECT treatments (all $p < 0.05$). No significant difference in serum BDNF levels was found at any of the indicated times between remitters and nonremitters, even when compared to baseline across the total samples (all $p > 0.05$, Fig. 2). In the linear mixed model, serum BDNF levels showed no significant main effects for group, time, or group-by-time interactions (all $p > 0.05$,

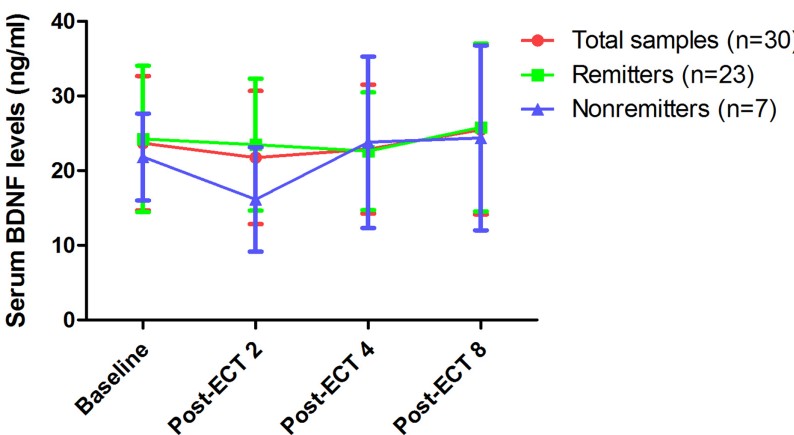

**Figure 2 Changes in serum BDNF levels following eight ECT treatments.** No significant difference was found at any of the indicated times when compared to baseline across the total sample, even among remitters and nonremitters ($p > 0.05$); no significant difference was found between remitters and nonremitters at any of the indicated times ($p > 0.05$). Abbreviations: BDNF, brain-derived neurotrophic factor; ECT, electroconvulsive therapy; TRD, treatment-refractory depression.

**Table 3 Comparison of serum BDNF levels, HAMD-17 scores and BPRS scores between remitters and nonremitters using linear mixed model analysis.**

| Variables | Group-by-time interaction | | Time main effect | | Group main effect | |
|---|---|---|---|---|---|---|
| | *F* | *P* | *F* | *P* | *F* | *P* |
| Serum BDNF levels | 1.2 | 0.321 | 1.6 | 0.214 | 0.9 | 0.355 |
| HAMD-17 scores | 3.7 | **0.019** | 2095.5 | **<0.001** | 5.0 | **0.028** |
| BPRS scores | 1.2 | 0.309 | 332.1 | **<0.001** | 1.6 | 0.208 |

**Notes:**
Bolded values are $p < 0.05$.
Abbreviations: BDNF, brain-derived neurotrophic factor; BPRS, the Brief Psychiatric Rating Scale; HAMD-17, the 17-item Hamilton Depression Rating Scale.

Table 3). Changes in HAMD-17 and BPRS scores following ECT treatment between remitters and nonremitters were also analysed using a linear mixed model (Table 3).

### Association between serum BDNF levels and illness severity

As reported in Table S1, there was no significant correlation between serum BDNF levels and illness severity as measured by HAMD-17 and BPRS.

## DISCUSSION

This study is, to our knowledge, the first to examine the relationship of serum BDNF levels and the rapid antidepressant effects of ECT with ketamine anaesthesia in Chinese patients with TRD. The main findings of the current study included the following: (1) there was no significant difference in baseline serum BDNF levels between healthy controls and patients with TRD; (2) following eight sessions of ECT with ketamine anaesthesia, all patients with TRD met the response criteria, and 76.7% remitted based on the criteria reported by *Riedel et al. (2010)*'s study; (3) ketamine used in ECT did not alter

serum BDNF levels compared to baseline, and serum BDNF levels between remitters and nonremitters was not significantly different; (4) there was no significant correlation between serum BDNF levels and improvement of depressive symptoms following eight sessions of ECT with ketamine anaesthesia.

In line with findings from early studies (*Fernandes et al., 2009*; *Huang et al., 2020*; *Maffioletti et al., 2019*; *Polyakova et al., 2015a*; *Ryan, Dunne & McLoughlin, 2018*), no significant difference was observed in serum BDNF levels between patients with TRD and healthy controls. However, other studies have reported that patients with depression exhibit reduced serum BDNF levels compared to healthy controls (*Allen et al., 2015*; *Karege et al., 2002*; *Kishi et al., 2017*; *Matrisciano et al., 2009*; *Nase et al., 2016*; *Rapinesi et al., 2015*; *Wolkowitz et al., 2011*; *Zheng et al., 2020*), suggesting that serum BDNF deficit might represent a potential biomarker in patients with depression. However, low serum BDNF levels were not specific to patients with depression, but were also observed in patients with schizophrenia and mania, and even in patients with acne vulgaris (*Karamustafalioglu et al., 2015*; *Li et al., 2016*; *Mikhael et al., 2019*; *Mora et al., 2019*). Taken together, blood BNDF level appears to have no value for diagnosis of depression in clinical practice.

After completing eight ECT treatments with ketamine as an aesthesia, 100% of patients with TRD responded in this study, and 76.7% met remission criteria, similar with the prior findings (*Huang et al., 2020*). *Huang et al. (2020)* found that 100% responded and 53.3% remitted after all TRD patients receiving eight ECT treatments with ketofol as aesthesia. *Zhong et al. (2016)* reported higher remission rates for patients with TRD receiving ECT with ketamine alone than propofol alone and ketofol, indicating that ketamine used alone as an aesthesia in ECT may be considered an optimal treatment in Chinese patients with TRD. Importantly, the findings of the current study also support the results of a meta-analysis of randomized controlled trials that ketamine used in ECT accelerates the antidepressive response and remission in depressed patients (*Ren et al., 2018*).

Consistent with findings of previous studies (*Allen et al., 2015*; *Huang et al., 2020*), serum BDNF levels did not significantly change during ECT with ketamine anaesthesia in this study. For example, *Huang et al. (2020)* found that ECT with ketofol anaesthesia did not alter serum BDNF levels, despite its rapidly decreasing depressive symptoms. However, findings on the effect of ECT with other anaesthesia types, such as methohexital or thiopental sodium, on serum BDNF levels are inconsistent (*Maffioletti et al., 2019*; *Vanicek et al., 2019b*). For example, *Vanicek et al. (2019b)* found a significant increase in serum BDNF levels after continuation ECT treatments with methohexital, while another study found that ECT with thiopental sodium did not alter serum BDNF levels (*Maffioletti et al., 2019*).

The current study also confirmed the findings of previous studies that the antidepressant effects of ECT with ketamine anaesthesia were not correlated with serum BDNF levels (*Allen et al., 2015*; *Huang et al., 2020*; *Kishi et al., 2017*). Similar to nonconvulsive electrotherapy (NET) using standard ECT technique but below seizure threshold, the change in serum BDNF levels in patients with TRD was not correlated with the antidepressant effects of NET (*Zheng et al., 2020*). A recent study concluded that

serum BDNF levels in patients with late-life unipolar depression cannot be considered an eligible biomarker for the antidepressant effects of ECT (*Van Zutphen et al., 2019*).

It is important to mention the following limitations of this study. The primary limitation of this study is the small sample size, partly accounting for the negative findings. Another limitation of the current study was the open-label design, limiting the interpretation of efficacy. For instance, *Carspecken et al. (2018)* found that ketamine used in ECT does not significantly improve depressive symptoms compared to methohexital used in ECT. Finally, the absence of follow-up visits after the entire treatment for this study limited our capacity to further examine how long the antidepressant effects of ECT with ketamine persists and whether increased BDNF concentrations requires a longer time to manifest. For instance, *Vanicek et al. (2019a)* found that peak serum BDNF levels were achieved one month after the final ECT treatment.

## CONCLUSION

In conclusion, this preliminary study indicated that serum BDNF levels do not appear to be a reliable biomarker to determine the antidepressant effects of ketamine as an anaesthesia in ECT for patients with TRD. Further studies with larger sample sizes are warranted to confirm these findings.

### Funding

This study was funded by the Science and Technology Planning Project of Guangdong Province (B2016109), the Science and Technology Planning Project of Liwan District of Guangzhou (202004034), and the Guangzhou Clinical Characteristic Technology Project (2019TS67). The funders had no role in study design, data collection and analysis, decision to publish, or preparation of the manuscript.

### Grant Disclosures

The following grant information was disclosed by the authors:
Science and Technology Planning Project of Guangdong Province: B2016109.
Science and Technology Planning Project of Liwan District of Guangzhou: 202004034.
Guangzhou Clinical Characteristic Technology Project: 2019TS67.

### Competing Interests

The authors declare that they have no competing interests.

### Author Contributions

- Wei Zheng analyzed the data, authored or reviewed drafts of the paper, and approved the final draft.
- Qiaomei Cen analyzed the data, prepared figures and/or tables, authored or reviewed drafts of the paper, and approved the final draft.
- Sha Nie performed the experiments, prepared figures and/or tables, and approved the final draft.

- Minyi Li performed the experiments, authored or reviewed drafts of the paper, and approved the final draft.
- Rong Zeng performed the experiments, authored or reviewed drafts of the paper, and approved the final draft.
- Sumiao Zhou analyzed the data, prepared figures and/or tables, and approved the final draft.
- Dongbin Cai analyzed the data, prepared figures and/or tables, and approved the final draft.
- Miaoling Jiang performed the experiments, authored or reviewed drafts of the paper, and approved the final draft.
- Xiong Huang conceived and designed the experiments, performed the experiments, authored or reviewed drafts of the paper, and approved the final draft.

### Human Ethics

The following information was supplied relating to ethical approvals (i.e., approving body and any reference numbers):

The ethics review board of the Affiliated Brain Hospital of Guangzhou Medical University approved this research (ethics number: (2013)020).

### Clinical Trial Registration

The following information was supplied regarding Clinical Trial registration:

The current study is an single-arm open label trial. Thus, the ID of the trial registry is not applicable.

### Data Availability

Raw data are available in the Supplemental Files.

### Supplemental Information

Supplemental information for this article can be found online at http://dx.doi.org/10.7717/peerj.10699#supplemental-information.

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
