# Peer review of "Serum BDNF levels and the antidepressant effects of electroconvulsive therapy with ketamine anaesthesia: a preliminary study"

_PeerJ, doi:10.7717/peerj.10699_

## Round 0.1 · original submission · Major Revisions

This is an interesting paper, however, there are still a lot of issues that need to be addressed. Please consider these comments and revise it accordingly.

Reviewer 1 ·

Basic reporting

None

Experimental design

None

Validity of the findings

None

Additional comments

This paper conducted a study that have investigated whether serum BDNF levels predict a rapid antidepressant response to ketamine used alone as an anesthesia during ECT in Chinese patients with TRD. There are several problems to be addressed:
(1) please notice the format of the abstract in the submit system
(2) In the result of “abstract”, please add the statistic values and P values.
(3) I checked the informed content and found that the protocol only associated with source of patients. Collecting the information from healthy volunteers also needs the informed content.
(4) A small sample size could lack enough statistic power to get the stable results. Authors should provide sample size estimation. I believe that authors can attain more healthy volunteers. The ratio (case: control=1:4) may have a higher statistic power.
(5) Line 115-117. Please provide the references to identify the feasibility of half-age method.
(6) Line 123. Please use “were” rather than “was”.
(7) Line 132. What is the meaning of “a single blood sample”?
(8) Line 157. Please use “baseline serum BDNF levels” rather than “serum BDNF levels”.
(9) Please specifically introduce the changes in HAMD-17 and BPRS scores following ECT treatment between remitters and non-remitters.
(10) I suggest that supplementary table S1 should be placed in the manuscript and Figure 1 should be placed in the supplementary materials.

Reviewer 2 ·

Basic reporting

No comment

Experimental design

No comment

Validity of the findings

No comment

Additional comments

1. The paper has some language issues. For example, line 6, a problematic sentence. Line 31, “major mental disorders” seems like a lay term in psychiatry, please make it clear.
2. The study has negative findings. One major limitation is the small sample size, 7 remitters were compared to 23 non-remitters and 30 TRD patients were compared to 30 healthy controls. Lack of statistical power should be a major reason for these negative findings. The conclusion seem also over-stated. Please consider to tone down it due to these limitations.
3. Line 58-79, the organization of the two paragraphs could be improved. The current review seems confused. I suggest the authors to review the biomarker roles of BDNF, either for antidepressant effect prediction or for disease severity, and review BDNF as a biomarker in ECT treatment, alone or combination with other anesthetics. For these reviews, it is important to emphasize more on positive findings on the biomarker roles of BDNF, because the authors hypothesized a positive role of BDNF in their studies. Inconsistent or negative findings on the biomarker roles could be moved to Discussion, because you have negative findings too!!
4. Line 98, there is no MDD or major depressive episode in ICD-10. Please check this.
5. Please clarify the clinical settings of patient recruitment, outpatient or inpatient?
6. Line 108, how patients and healthy controls were matched?
7. Line 113, in addition to the treatment sessions, please specify how long was the treatment delivered?
8. Line 121, clinical assessment. Safety outcomes are also important for clinical trials, but the authors only had efficacy outcomes.
9. In the supplementary study protocol, there were three treatment arms, but in this report, there was only one arm, ECT + Ketamine. Please specify why the two control groups were not included into the analysis.
10. Figure 2, please check the data of non-remitters. Scores of non-remitters and remitters are very similar to 6.
11. For outcome assessment, the authors defined treatment response, but no corresponding results reported.

---

## Round 0.2 · Minor Revisions

Please further specify the way of selecting healthy controls, because this determines the statistical methods.

Reviewer 1 ·

Basic reporting

None

Experimental design

None

Validity of the findings

None

Additional comments

Thank you for your revision. However, when I read your reply,I found that you told reviewers that patients and healthy controls were matched one by one. So Did you use individual matching? If so,student’s t-test, chi-square test did not good choices. Pair t test and McNemar chi-square test are better ways to analyze the Paired-design data.

---

## Round 0.3 · accepted · Accept

Thank you for the detailed revisions.

Reviewer 1 ·

Basic reporting

None

Experimental design

None

Validity of the findings

None

Additional comments

None